**nature** COMMUNICATIONS

# Temperature-dependent kinetic pathways of heterogeneous ice nucleation competing between classical and non-classical nucleation

Chu Li [1], Zhuo Liu[1,2], Eshani C. Goonetilleke [1] & Xuhui Huang [1✉]

Ice nucleation on the surface plays a vital role in diverse areas, ranging from physics and cryobiology to atmospheric science. Compared to ice nucleation in the bulk, the water-surface interactions present in heterogeneous ice nucleation complicate the nucleation process, making heterogeneous ice nucleation less comprehended, especially the relationship between the kinetics and the structures of the critical ice nucleus. Here we combine Markov State Models and transition path theory to elucidate the ensemble pathways of heterogeneous ice nucleation. Our Markov State Models reveal that the classical one-step and non-classical two-step nucleation pathways can surprisingly co-exist with comparable fluxes at $T = 230$ K. Interestingly, we find that the disordered mixing of rhombic and hexagonal ice leads to a favorable configurational entropy that stabilizes the critical nucleus, facilitating the non-classical pathway. In contrast, the favorable energetics promotes the formation of hexagonal ice, resulting in the classical pathway. Furthermore, we discover that, at elevated temperatures, the nucleation process prefers to proceed via the classical pathway, as opposed to the non-classical pathway, since the potential energy contributions override the configurational entropy compensation. This study provides insights into the mechanisms of heterogeneous ice nucleation and sheds light on the rational designs to control crystallization processes.

[1] Department of Chemistry, Center of Systems Biology and Human Health, State Key Laboratory of Molecular Neuroscience, The Hong Kong University of Science and Technology, Kowloon, Hong Kong. [2] Institute for Advanced Study, The Hong Kong University of Science and Technology, Kowloon, Hong Kong. ✉email: xuhuihuang@ust.hk

Crystal nucleation from melt[1], where the nucleus of ordered structures emerges from a liquid, significantly influences science and technology, ranging from atmospheric science to the design of functional materials[2–7]. As a result, crystal nucleation has a profound impact on our daily life. Therefore, understanding the thermodynamics and kinetics of the nucleation process is imperative and has become a topic under intense study[8–12]. Classical nucleation theory (CNT) provides a phenomenological description for nucleation, based on the thermodynamic framework of overcoming a single free energy barrier that is attributed to the balance between the energy gained from forming a nucleus and the energy lost from the creation of interfaces[13]. However, recent experiments[3,11,14–17] and simulations[18–21] have shown that CNT is often insufficient to describe complex nucleation processes. For this reason, non-classical nucleation theories based on kinetic perspectives (e.g., the Ostwald step rule[22,23]) have been proposed, in which the nucleation process consists of two or even more steps separated by multiple free energy barriers[1,24]. Non-classical crystal nucleation has been reported in a wide range of systems, including the crystallization of nanoparticles[14,25], proteins[11,17], and ionic compounds[15,16,19]. Nevertheless, the atomistic mechanisms and comprehensive elucidation of the structural evolutions during nucleus formation in the non-classical two-step nucleation pathways are not well known; and it remains challenging for experimental techniques to elucidate the kinetic pathways and reveal intermediate states for the crystal nucleation processes.

Heterogeneous ice nucleation (HIN), i.e., the formation of ice nuclei on a foreign surface, is one of the most ubiquitous crystal nucleation processes on earth[1], shaping our world from climate mediation to glacier formation[6,7]. The unique properties of the hydrogen bond network in water also make ice nucleation an archetypal and intriguing system to study[26–31], drawing vast attention from physicists, chemists, and material scientists[32–35]. Although various experimental techniques have been adopted to probe ice nucleation in a broad spatiotemporal scale[36–39], it remains challenging for experiments to characterize ice nucleation at characteristic resolutions, i.e., temporal (pico- to microsecond) and spatial (atomistic). Alternatively, molecular dynamics (MD) simulations, a powerful tool with high temporal (up to femtoseconds) and spatial (atomistic) resolutions, have been extensively applied to study the dynamics of ice formation[40–48]. Interestingly, previous studies[40] have found that, even though homogeneous ice nucleation follows a one-step nucleation pathway, the formation of stacking-disordered ice nuclei is preferred due to a favorable entropic stabilization. However, it remains unclear whether the mechanism of entropic stabilization exists in HIN, and how the interactions from the surface interfere with the nucleation process. Previous studies[46,49] have found that a single-barrier CNT is not sufficient to describe HIN, in which pre-nucleation polymorphic structures found at an early stage suggest a two-step nucleation process. However, previous simulation studies of ice nucleation on the surface mostly focus on individual simulation trajectories[44], or one reaction pathway[50,51], which may not be sufficient to systematically identify an ensemble of highly parallel kinetic nucleation pathways. Furthermore, it is challenging to reveal the critical nucleus and the intermediate states, which are crucial when characterizing the mechanisms for either the classical one-step or non-classical multi-step nucleation pathways.

Markov State Models (MSMs)[52–58] constructed from many short MD simulations provide a promising approach to elucidate the complicated kinetic pathways of ice nucleation. In an MSM, the continuous dynamics are modeled as Markovian transitions among metastable states at discrete time intervals[59–66]. Moreover, MSMs have become a popular and powerful tool to simulate biomolecular dynamics, such as RNA[63] and protein[67–72] folding, ligand-receptor binding[73–75], and protein allostery[76]. In addition, researchers, including ourselves, have previously applied MSMs to clarify the dominant self-assembly pathways of nanoparticles[77,78] and lipids[79], where MSMs can automatically identify intermediate states and calculate their respective thermodynamic and kinetic properties at equilibrium. To our knowledge, MSMs have not been previously used to elucidate the kinetics of ice nucleation.

In this work, we employed MSMs and transition path theory[80–82] (TPT) to elucidate the ensemble of kinetic pathways of HIN simulated by MD simulations on a wurtzite-structured surface. Strikingly, our MSMs show that when water is supercooled to 230 K, ice nucleation can proceed via either the classical one-step or the non-classical two-step nucleation pathway, in which the non-classical pathway differentiates remarkably from the classical pathway by containing two activation steps. Moreover, compared to the classical pathway, the increase in configurational entropy from the disordered mixing of rhombic and hexagonal ice stabilizes the critical nucleus in the non-classical pathway, making both pathways accessible with comparable flux. Furthermore, by elevating the temperature, we discover that the nucleation process shifts significantly towards the classical pathway, mainly because the potential energy difference, which favors the classical pathway, prevails over the configurational entropy compensation.

## Results and discussions

**Collective variables to describe the kinetics of ice nucleation.** Ice nucleation on the surface is a heterogenous and collective process that involves thousands of molecules; therefore, it is important to identify proper collective variables (CVs) that can precisely describe the dynamics of HIN simulated from MD (simulation details are provided in Methods and Supplementary Methods 1). We adopted the Spectral-oASIS approach[83,84] to automatically choose CVs that can describe the slowest dynamics relevant to the kinetics of HIN from a pool of 18 candidate CVs that were identified based on physical intuition. The Spectral-oASIS chose five representative CVs by utilizing the Nyström method to approximately reconstruct the time-lagged correlation matrix of all candidate CVs using only a subset of the CVs. As listed in Supplementary Table 1, the initial candidate CVs describe various structural and geometric features of the ice nucleus. The structural features (including the molecular numbers of hexagonal and rhombic ice) were obtained by using average bond order parameters[85] to distinguish the local structures of each water molecule (see Methods and Supplementary Methods 2 for more detail), and the spherical parameter[77] was used to monitor the geometric evolution of the ice nucleus. As illustrated in Fig. 1a and Supplementary Fig. 1, the hexagonal ice forms three hydrogen bonds with its intralayer neighbors and one with its interlayer neighbor; whereas the rhombic ice mainly forms four hydrogen bonds with its intralayer neighbors, despite some neighbors being buckled as opposed to being totally flat[49]. Structures[41,49] similar to rhombic ice have been reported on different surfaces, even at room temperature[41]. The molecular numbers of hexagonal and rhombic ice are further divided into different layers, as displayed by Fig. 1b, due to the characteristics of layer-by-layer ice crystallization on the surface.

The five optimal CVs selected by the Spectral-oASIS account for the numbers of ice molecules in the largest ice nucleus, rhombic ice in the largest ice nucleus, and hexagonal ice in the 2nd, 3rd, and upper layers of the largest ice nucleus (see Supplementary Fig. 2). Furthermore, Fig. 1c shows that out of the

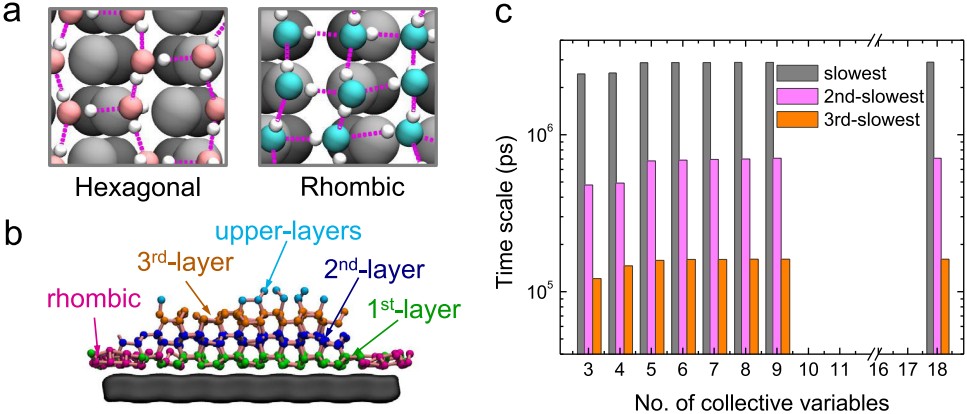

**Fig. 1 Selection of representative collective variables (CVs). a** Illustration of the structures of hexagonal (left) and rhombic (right) ice. The dashed lines represent the water-water hydrogen bonds. Rhombic ice tends to form four hydrogen bonds with its intralayer neighbors, forming a network that is slightly buckled, while hexagonal ice usually forms a dangling hydrogen bond with an interlayer neighbor. The oxygen atoms in hexagonal and rhombic ice, hydrogen atoms, and surface atoms are shown as pink, cyan, white, and black spheres, respectively. **b** A typical definition of structural information in the system, where hexagonal and rhombic ice is categorized into different layers. **c** Timescales for the dynamics of the system with various numbers of CVs. With five selected CVs, the dynamical information from the complete set of 18 CVs can be reconstructed with respect to the slowest three dynamics with minimal reconstruction errors.

five optimal CVs, the slowest three timescales approach those of the complete 18 CVs, indicating that the reconstructed time-lagged correlation matrix with the selected CVs is sufficient to maintain the slowest dynamical information with minimal errors. The slowest timescales are strongly correlated with ice formation in different layers, as discussed in Supplementary Methods 3.1. Interestingly, the spherical parameter was not selected, most likely due to the ice nucleus being relatively flat. This implies that the spherical parameter cannot properly describe the kinetics of HIN in our system. Instead, the parameters relating to the size of the largest ice nucleus can describe the kinetics of ice nucleation. This agrees with a previous study[51], which showed that the size of the ice nucleus is essential to describe ice nucleation. However, due to the presence of rhombic ice and the layer-by-layer characteristic of ice crystallization, our system needs more CVs to accurately reconstruct the full dynamics shown in Fig. 1c, despite all being size-related.

**MSMs reveal competitive kinetic pathways for classical and non-classical nucleation at 230 K.** Based on the five selected CVs, we constructed MSMs following a two-step procedure using the MSMbuilder Packages[86] (see Methods and Supplementary Methods 3 for more detail): (i) Based on the *k*-centers clustering algorithm[87], we grouped the MD configurations with the CVs into 1000 microstates, chosen by variational cross-validation with a generalized matrix Rayleigh quotient[88] (GMRQ). Then, the MSMs were validated by implied timescale analysis and the Chapman–Kolmogorov test[89] based on the microstate level (see Supplementary Fig. 3 and 4 for more detail), implying that the MSMs reach Markovian at the microstate level. (ii) We then further grouped the 1000 microstates into eight macro-states to better visualize the nucleation kinetics using the Robust Perron Cluster Clustering Analysis (PCCA + ) algorithm[62]. The macro-states from MSMs represent metastable regions (i.e., free energy basins), and are separated by free energy barriers (i.e., transition states).

Strikingly, our MSMs revealed that both the classical one-step and non-classical two-step HIN pathways coexist at $T = 230$ K. Representative kinetic trajectories corresponding to either one or two activation steps are displayed in Fig. 2a, b, respectively (see Methods and Supplementary Methods 5 for more detail on generating these kinetic trajectories using Markov Chain Monte

Carlo simulations based on the MSMs; representative original MD trajectories with one-step and two-step characteristics are also presented in Supplementary Fig. 5 and 6, respectively). This clearly indicates that HIN can proceed via either the one-step or two-step pathway, with each activation corresponding to the crossover of a free energy barrier as sketched in the subsets. To further characterize the flux of these two nucleation pathways, we applied transition path theory[80–82,90], which allowed us to reveal the ensemble of transition paths[68]. Fig. 2c displays the top four fluxes that assemble the transitions between pairs of macro-states, whose representative configurations are snap-shotted in the front view, together with their [101] view in Fig. 2d (see Supplementary Fig. 7 for complete fluxes). Fig. 2c shows that the rhombic ice and hexagonal ice compete to form during the early stage of nucleation, causing the flux to bifurcate into two distinguished pathways. Specifically, the flux that transitions from macro-states I → II → IV (gray arrow), suggests a direct formation of hexagonal ice with little involvement of rhombic ice, which corresponds to the classical one-step nucleation pathway. In the other direction, the fluxes that transition from macro-states I → II → III (purple, orange, and red arrows) suggest a significant increase in rhombic ice and hexagonal ice at the first step. The rhombic ice then transforms back into hexagonal ice via macro-states III → IV/V as the second step, as shown by the structural evolutions of the nucleus along with the three fluxes (i.e., purple, orange, and red). These fluxes (including fluxes in purple, red, and orange) belong to the non-classical two-step nucleation pathway[11,24] since they all involve the intermediate development of characteristic pre-nucleation structures, i.e., a mixture of rhombic and hexagonal ice, which is different from the classical description of the direct formation of hexagonal ice. After the conversion of rhombic ice into hexagonal ice, the fluxes coalesce and enter the growth stage, as shown by the further growth of hexagonal ice.

Furthermore, the mean first passage times (MFPTs) for each transition between the two macro-states are reported in Supplementary Fig. 7 (see Methods for more detail). Interestingly, the MFPT to proceed along the non-classical pathway is comparable but a bit longer than that of the classical pathways, consistent with the fluxes along the two competitive pathways. More importantly, the heterogeneous ice nucleation rate can be estimated from the MFPTs as $J_{het} = 3.5 \times 10^{30 \pm 1}$ s$^{-1}$m$^{-3}$ for our system at $T = 230$ K (see Supplementary Methods 5), which is

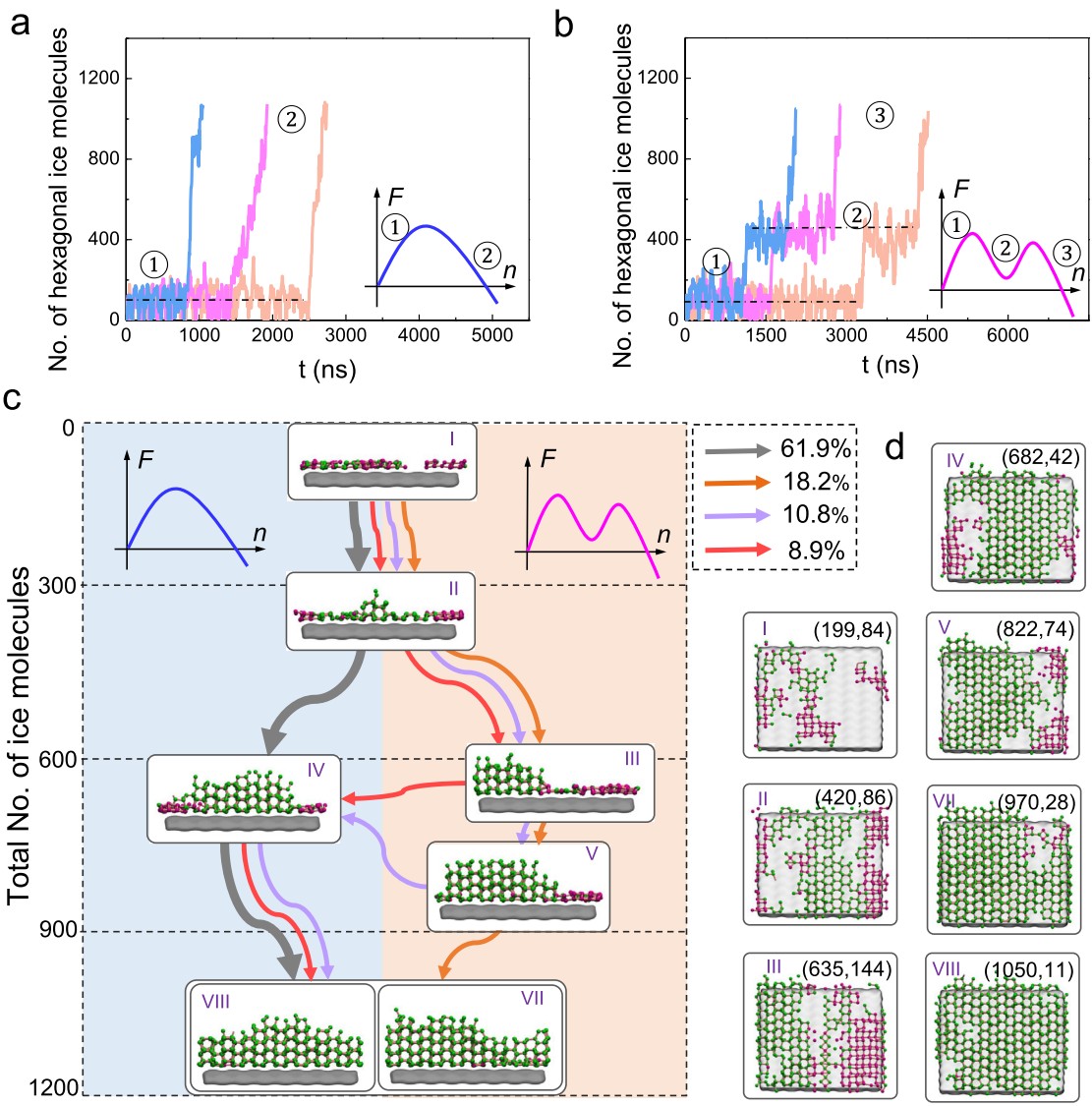

**Fig. 2 The kinetics of HIN at $T = 230$ K.** Typical trajectories of the classical one-step nucleation pathway (**a**) and the non-classical two-step nucleation pathway (**b**), which are characterized by one and two activations, respectively. The subsets sketch the corresponding free energy landscapes for the one-step and two-step nucleation pathways. **c** The kinetic pathways of HIN: coexistence of the classical and non-classical nucleation pathways. For each macro-state, a representative front view and corresponding [101] view (**d**) of the ice configuration is shown, where hexagonal and rhombic ice molecules are represented by green and purple spheres, respectively. The connections between each pair of macro-states form a network that depicts the kinetics of the transitions. The arrows represent the directions of the net flux for each transition. Four major paths with a net flux greater than 5% are identified with different colors. The thickness of the arrows indicates the flux of the nucleation process between each pair of macro-states. The path following the gray arrows depicts the classical pathway, while the paths following the red, purple, and orange arrows represent the typical non-classical nucleation pathway. Along the non-classical pathway, a significant amount of rhombic ice forms intermediately and disappears as nucleation proceeds. In (**d**), the numbers in the bracket represent the average number of molecules of total ice and rhombic ice in the largest ice nucleus in each macro-state. The surface is shown as a cartooned gray surface. Macro-states VII and VIII are combined as the final state as they have minor differences and the system has entered the growth stage.

orders of magnitude faster than the rate for homogeneous ice nucleation ($J_{\text{hom}} \approx 10^{6 \pm 1}$ s$^{-1}$m$^{-3}$)[47]. We also notice that a previous study[50] reported a fast heterogeneous ice nucleation rate of $J_{\text{het}} = 10^{26 \pm 2}$ s$^{-1}$m$^{-3}$ on a Kaolinite surface using the same water model at 230 K. These underline the relevance of studying heterogeneous ice nucleation as the major mode in competition with homogeneous ice nucleation when water is supercooled to $T = 230$ K.

**Entropy stabilization of the critical nucleus for the non-classical nucleation pathway via the disordered mixing of rhombic and hexagonal ice.** The free energy landscape where the

nucleation pathway bifurcates with the transition states (TSs) at the saddle points which govern the transitions, is illustrated in Fig. 3a. Specifically, TS I in Fig. 3c corresponds to the critical ice nucleus for the classical one-step pathway; and TS II in Fig. 3b corresponds to the critical ice nucleus for the 1$^{\text{st}}$ barrier of the non-classical two-step pathway (see Methods for the identification of the TSs using TPT). Interestingly, the MFPTs for the transition from macro-states I to III are shorter than that from macro-states I to IV (i.e., 1.34 vs. 1.65 μs). This suggests that the 1$^{\text{st}}$ free energy barrier along the non-classical pathway is smaller than that for the classical pathway. However, the difference in average potential energy (including the potential energy within the ice nucleus, ice-surface interaction energy, and ice-liquid

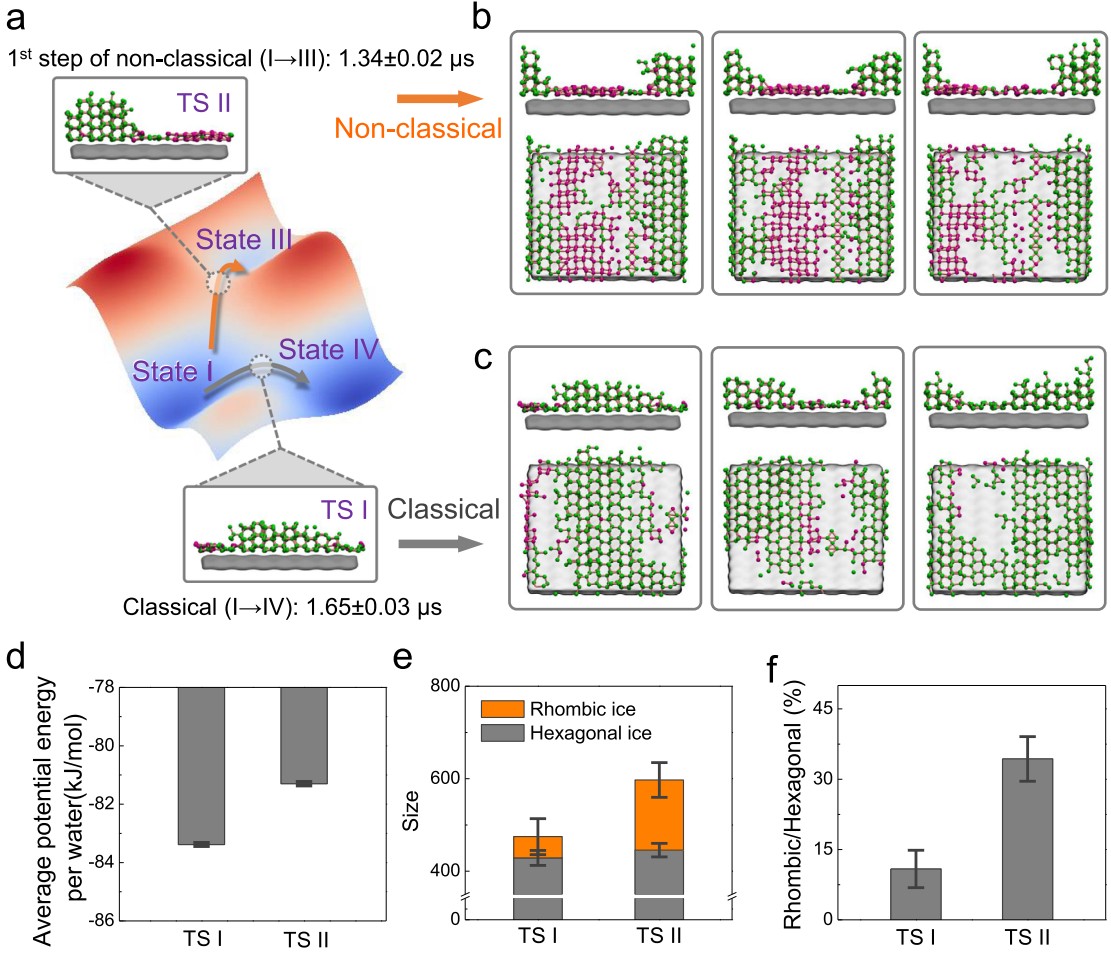

**Fig. 3 Critical nuclei along the two nucleation pathways.** The critical nuclei characterized by the transition states (TSs) are obtained by two independent transition path theory (TPT) analyses, in which the source is localized in macro-state I, where the separation of the two paths initiates, and the sinks are selected independently at macro-states III and IV, where the separation is finished. **a** Schematic of the free energy landscape for the formation of the ice nucleus proceeding via two separate pathways. Macro-states I to IV (gray) represents the crossover of the free energy barrier along the classical ice nucleation pathway with the direct formation of hexagonal ice. Macro-states I to III (orange) represents the crossover of the 1st free energy barrier along the non-classical two-step ice nucleation pathway, where the ice nucleus is comprised of a mixture of rhombic and hexagonal ice. **b** Representative configurations (top: front view; bottom: [101] view) of the critical nucleus for TS II. **c** Representative configurations (top: front view; bottom: [101] view) of the critical nucleus for TS I. **d** Comparison of the two TSs with respect to the average potential energy per water molecule of the critical nucleus, where the error bars represent the standard errors with samples obtained by bootstrapping. **e** Comparison of the two TSs with respect to the amount of rhombic and hexagonal ice in the critical nucleus, the error bars represent the standard errors. **f** Comparison of the two TSs with respect to the disorderliness (defined as the ratio of the number of rhombic and hexagonal ice) of the nucleus, the error bars represent the standard errors.

water interaction energy) for each water molecule between the critical nuclei in TS I and II is approximately 2.08 kJ mol$^{-1}$ (see Fig. 3d). Considering the size of the critical nucleus, this average potential energy difference could result in a free energy difference that favors the classical pathway. We found that this difference can be compensated for by the entropy stabilization that results from the disordered mixing of rhombic and hexagonal ice on the surface. Specifically, Fig. 3e, f show a larger fraction of rhombic ice in TS II, which contributes to a greater number of nucleus configurations compared with TS I. This increased number of configurations leads to a favorable configurational entropy, stabilizing TS II, and alleviating the free energy barrier along the non-classical pathway.

The configurational entropy stabilization of the nucleus via the disordered mixing of rhombic and hexagonal ice makes the non-classical nucleation pathway accessible. Similar mechanisms occur in homogeneous ice nucleation[40], where the critical ice nucleus is stabilized by the increase in entropy from the stacking-disordered cubic and hexagonal ice. However, unlike

homogeneous ice nucleation, the classical HIN pathway, which forms mainly hexagonal ice, is also accessible with comparable flux due to the favorable potential energy for the critical nucleus (TS I). This can be partially attributed to the surface, which has stronger interactions with the first layer of hexagonal ice than rhombic ice, as shown in Supplementary Fig. 8. Another factor that might lead to the lower potential energy for TS I can be the noticeably smaller amount of ice at the liquid-ice interface compared to TS II (taller ice nucleus structures in TS II than those in TS I, see Fig. 3b, c), as ice at the liquid-ice interface is generally less stable with higher potential energy[91]. In addition, for the non-classical HIN pathway, the formation of a disordered mixture of rhombic and hexagonal ice facilitates the system to overcome the first free energy barrier, after which the rhombic ice is further converted into hexagonal ice via transitioning from states III → IV or III → V (at around 0.59 and 1.17 μs, respectively), as shown in Fig. 2c. The conversion is likely attributed to an attenuated mixing fraction as the ice nucleus grows with the formation of hexagonal ice (since rhombic ice

mostly forms in the first layer, and the ice nucleus above the first layer is mainly hexagonal ice), and a reduction in potential energy when the rhombic ice and the liquid above the rhombic ice are converted into hexagonal ice.

**Elevated temperatures favor the classical nucleation pathway with the formation of hexagonal ice.** To investigate the temperature dependence of the HIN process, we constructed MSMs at elevated temperatures of 240 and 250 K; and in order to compare the kinetics with HIN at 230 K, we utilized the same state-decomposition as the microstate-MSM at $T = 230$ K (see Methods and Supplementary Fig. 9 and 10 for more detail on MSMs construction and validation). Our results show that as temperature increases, the nucleation at $T = 240$ and 250 K shifts towards the classical one-step nucleation pathway via states I → II → IV → VII/VIII (Fig. 4a, b, complete MSMs are presented in Supplementary Fig. 11 and 12, respectively), as revealed by the fluxes of the two pathways at different temperatures in Fig. 4c. Furthermore, the rise in MFPTs in Supplementary Fig. 7, 11, and

12, show that the nucleation process takes longer to reach states IV from state I when proceeding via the one-step pathway, implying that a larger energy barrier exists along this nucleation pathway. This qualitatively agrees with CNT and previous studies[29,92], which showed that the free energy barrier for ice nucleation rises as $T$ increases.

Temperature affects the thermal fluctuations of water molecules, and consequently, the nucleation pathway. As previously discussed, the free energy barrier of the critical nucleus for the non-classical pathway is comparable to that of the classical pathway at $T = 230$ K, and this results from the balance between the potential energy and entropy that arises from the disordered mixing of different ice structures. However, the balance between these two contributions is sensitive and can be broken by other factors, including temperature. At higher $T$, the total entropy of the system increases and tends to drive the system into liquid water, leading to an increase in the free energy barrier along both pathways simultaneously. The elevated free energy barrier will correspond to a larger critical nucleus. For example, in our system, the size of the critical nucleus for the classical pathway

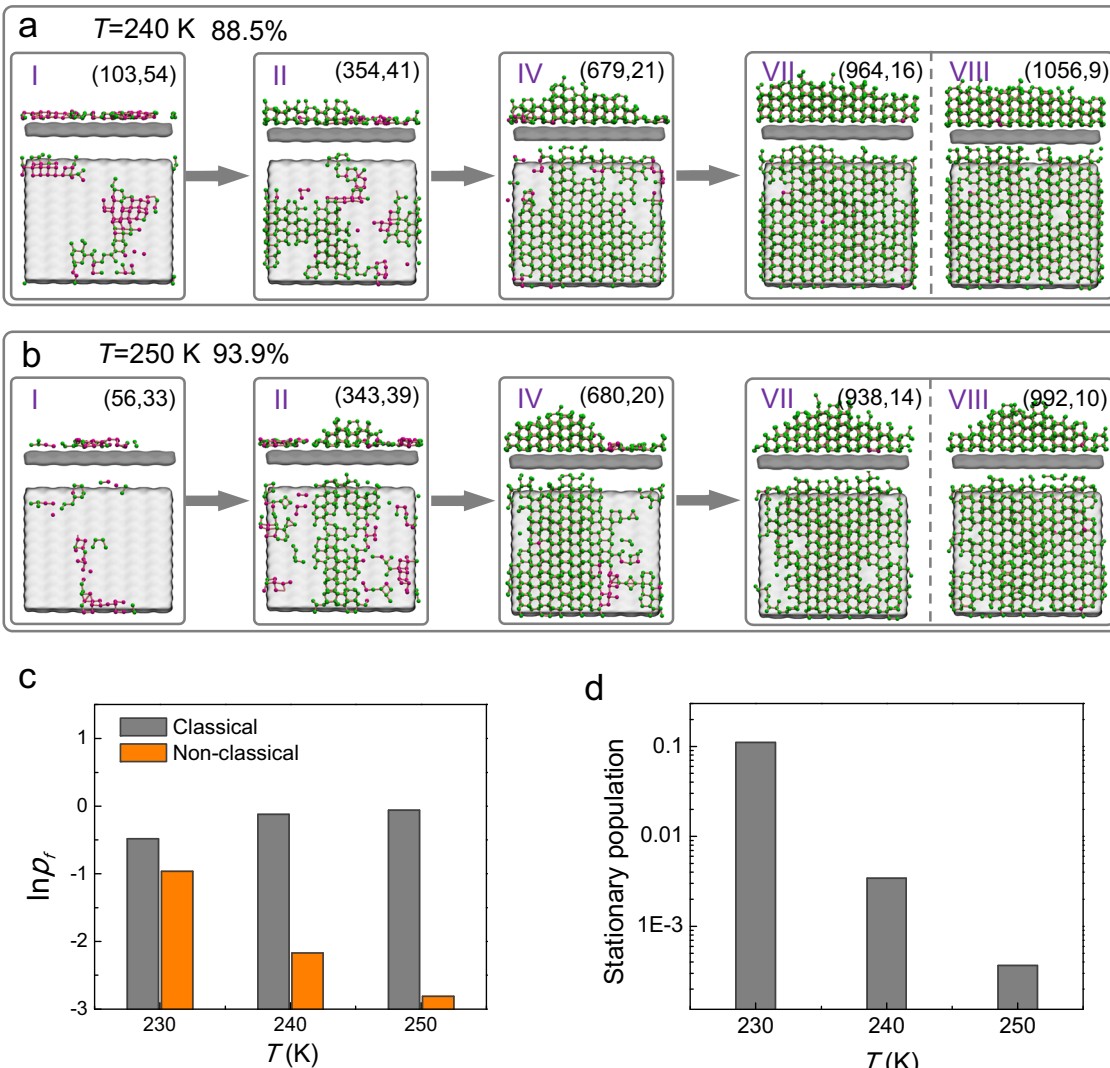

**Fig. 4 Elevated temperature promotes HIN to shift to the classical pathway. a** Major pathway at $T = 240$ K. **b** Major pathway at $T = 250$ K. The classical pathway dominates in these two cases. Each snapshot presents the typical average configuration of the corresponding macro-state in both the front view and the [101] view, in which purple and green spheres represent the rhombic and hexagonal ice molecules, respectively. **c** Comparisons of fluxes for the classical and non-classical pathways at different temperatures. **d** The stationary population of macro-state III by which all the non-classical fluxes pass at different temperatures, which are based on microstate-MSMs.

also increases with temperature (i.e., the total number of ice molecules = 475, 593, and 665 when $T$ = 230, 240, and 250 K, respectively), which agrees with CNT and previous work[29]. As a result, the potential energy difference between the two TSs along the two pathways will increase with nucleus size. In terms of configurational entropy, we anticipate that the difference between the two pathways may not increase much since the rhombic ice mainly forms in the first layer, and the increase in critical nucleus size (i.e., from the formation of mainly hexagonal ice since the first layer has been fully occupied) will not increase the configurational entropy enough to compensate for the potential energy difference. Therefore, the potential energy difference overrides the configurational entropy compensation, making the classical nucleation pathway more accessible than the non-classical pathway. As a result, during the early stage before the bifurcation of nucleation flux, the nucleation process becomes more likely to proceed via the classical pathway, with the dominant formation of hexagonal ice, which can also be verified by the dramatic decrease of the stationary population of macrostate III based on microstate-MSMs in Fig. 4d, by which all the non-classical pathways pass.

MSMs combined with the TPT analysis offer a powerful framework to study the structural evolutions of ice nucleation and might be extended to other crystal nucleation processes that are challenging to study when using direct or unbiased MD simulations. Furthermore, MSMs allow us to identify hidden transition states of the disordered ice mixture and compare multiple parallel pathways, uncovering the underlying mechanisms of the co-existence of the classical and non-classical HIN pathways. Different from homogeneous ice nucleation, which has one-step and one dominant pathway[40], the presence of a foreign surface in HIN can, under certain conditions, give rise to parallel one-step and two-step pathways. The critical ice nuclei along the two pathways are stabilized by contrasting energetics, showing a delicate competition of the driving forces between intrinsic entropy stabilization and water-surface interactions. In addition, the temperature-dependent kinetics makes it feasible to tune the nucleation pathway, offering a possible explanation for the contrasting scenarios of both the classical and non-classical crystal nucleation observed in experiments.

In conclusion, we constructed MSMs to elucidate the kinetic pathways of HIN. Using MSMs along with TPT analysis, we were able to illustrate the competitive kinetics of the classical one-step and the non-classical two-step ice nucleation pathways, and uncover the underlying mechanism of comparable flux at $T$ = 230 K. In particular, we show that the configurational entropy stabilization of the critical nucleus via the disordered mixing of rhombic and hexagonal ice makes the non-classical nucleation pathway as accessible as the classical pathway, whose critical nucleus mainly consists of potential energy-favored hexagonal ice. Furthermore, we compared the major pathways by altering the temperature; and found that as temperature increases, the dominant nucleation flux shifts to the classical nucleation pathway. This study shows that MSMs are promising tools when investigating crystal nucleation, and sheds light on the mechanisms of non-classical nucleation via entropy stabilization. Moreover, by altering temperatures, we are able to achieve modified nucleation kinetics, paving the way to control the crystallization process.

## Methods

**System setup and all-atom MD simulations**. The all-atom $NVT$ MD simulations were performed using LAMMPS packages[93] with a time step of 2 fs. The simulation system consisted of 4940 water molecules sitting on top of a wurtzite-structured surface in a periodic box with dimensions: $x$ = 5.43 nm, $y$ = 5.89 nm, and $z$ = 7.0 nm, as shown in Supplementary Fig. 1a. The TIP4P/Ice[94] water model was

employed to simulate water molecules, and the surface atoms were fixed in the wurtzite lattice under unit cell parameters of $a$ = 4.519 Å and $c$ = 7.357 Å. The wurtzite structure is shared by various materials such as AgI[43] and the unit cell parameters were chosen to match the hexagonal ice to minimize strain energy due to lattice mismatch. The water-surface interactions were simulated by the Lennard-Jones (LJ) potential, which was cut off at 1.0 nm. Initially, unbiased all-atom simulations were performed at $T$ = 230 K and $T$ = 240 K. Seeded unbiased simulations were shot for $T$ = 250 K, with original seeds from $T$ = 240 K. (see Supplementary Methods 1 for more detail).

**Ice detection**. To characterize the local structures of a water molecule, we employed the average bond order parameter[85], which calculates the local order of an atom with its neighbors as follows:

$$\bar{q}_l(i) = \left( \frac{4\pi}{2l+1} \sum_{m=-l}^{m=l} \left| \frac{1}{N_i+1} \sum_{j=0}^{N_i} q_{l,m}(j) \right|^2 \right)^{1/2}, \tag{1}$$

where $N_i$ is the number of atom $i$'s closest neighbors $j$, and $q_{l,m}(j) = \frac{1}{N_j} \sum_{k=1}^{N_j} Y_{l,m}(\mathbf{r}_j - \mathbf{r}_k)$ with $Y_{l,m}, \mathbf{r}_j$, and $\mathbf{r}_k$ being the spherical harmonics, position vectors of $j$ and its neighbor $k$, respectively. For bulk water, $\bar{q}_6 = 0.45$ can differentiate hexagonal ice[49] from liquid water with $N_i$ = 4. While for water at the interface, $N_i$ = 3 and water with ($\bar{q}_6 > 0.5$, $\bar{q}_4 < 0.6$) and ($\bar{q}_4 > 0.6$) are characterized as hexagonal ice and rhombic ice[49], respectively, whose corresponding structures are presented in Fig. 1a. (Supplementary Methods 2 for more detail).

**Construction and validation of MSMs**. We constructed MSMs to study the kinetics of HIN at $T$ = 230 K through the following procedures (see Supplementary Methods 3 for more detail): (i) We utilized Spectral-oASIS to automatically selected five CVs that could precisely describe the kinetics of HIN, from a pool of 18 candidate CVs obtained based on physical intuition (as listed in Supplementary Table 1); (ii) We grouped the MD configurations with the five optimal CVs into 1000 microstates based on the $k$-centers clustering algorithm[87]. The hyperparameter for clustering, i.e., the number of microstates, was determined by conducting variational cross-validation with the GMRQ[88], as illustrated in Supplementary Fig. 14; (iii) We validated our microstate MSMs by implied timescale analysis and Chapman–Kolmogorov test[89], as shown in Supplementary Fig. 3 and 4. (iv) To better visualize the nucleation kinetics, we further grouped the 1000 microstates into eight macro-states based on the PCCA+ algorithm[62], as illustrated in Supplementary Fig. 3a.

MSMs were also constructed to study HIN at $T$ = 240 and 250 K (see Supplementary Methods 4). To better compare the difference in flux at different temperatures, we projected each frame into the same microstates and macro-states obtained from $T$ = 230 K. The MSMs were also validated by the implied timescales and Chapman–Kolmogorov tests, as shown in Supplementary Fig. 9 and 10.

**Millisecond trajectories and mean first passage time**. We also used Markov Chain Monte Carlo simulations based on the transition probability matrix of the 1000-microstate MSM to generate ten independent 2 ms-long trajectories (see Supplementary Methods 5 for more detail). From these generated trajectories, we were able to recover the one-step and two-step nucleation pathways for $T$ = 230 K, as shown in Fig. 2a–b. The generated trajectories were also used to calculate the MFPTs of the transitions between pairs of states.

**Flux analysis and identification of critical ice nucleus**. The flux of the nucleation pathways were analysed based on a mathematical framework of TPT[80–82,90]. We computed the ensembled flux of the nucleation pathways along with their relative probabilities at the microstate level. For visualization purposes, the net flux of the transitions was assembled into coarse-grained macro-states[68]. To identify the critical ice nuclei along the classical and non-classical pathways, the saddle points along the two pathways were obtained using two independent TPT analyses based on the committor analysis at the microstate level (see Supplementary Methods 6 for more detail).

## Data availability

The MD trajectories for $T$ = 230, 240, and 250 K used to build the MSMs have been deposited in the Open Science Framework[95].

## Code availability

The MD simulations are carried out by LAMMPS packages[93], and the MSMs are constructed using MSMbuilder Packages[86]. All the data analysis performed in this study has been done by the MSMbuilder[86].

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

## Acknowledgements

This work was supported by Hong Kong Research Grant Council [grant numbers: 16302214, 16307718, 16303919, N_HKUST635/20, T13-605/18-W, AoE/M-09/12 and AoE/P-705/16]; Innovation and Technology Commission [grant numbers: ITCPD/17-9 and ITC-CNERC14SC01]; This research made use of the computing resources of the Supercomputing Laboratory at King Abdullah University of Science and Technology and the X-GPU cluster supported by the Hong Kong Research Grant Council Collaborative Research Fund: C6021-19EF.

## Author contributions

C. L. and X. H. designed the research. C. L. performed the simulations and data analysis. C. L. and X. H. interpreted and analyzed the results. C. L., Z. L., E. C. G., and X. H. discussed the results and contributed to writing the paper.

## Competing interests

The authors declare no competing interests.
