## [Peer Review File · Nature Communications]

REVIEWER COMMENTS

Reviewer #1 (Remarks to the Author):

This paper provides a detailed investigation on pathways of heterogeneous ice nucleation in undercooled water as revealed by atomistic computer simulations. The authors implement a very powerful technique for the analysis of activated events that is well known in the Bio community under the name of 'Markov State Model', but does not appear to have been applied extensively to the study of heterogeneous nucleation. The technique allows to identify relevant collective variables and to partition phase space systematically into macroscopic states. With a combination of Transition State sampling, it provides the fluxes from one state to the other, allowing for a detailed but amenable characterization of nucleation pathways. The authors have made a very large computational effort, and perform the numerical analysis of results with great expertise. Furthermore, the manuscript is well written, and straight to the author's main claim: namely, that there are two different significant pathways for heterogeneous nucleation in ice. Unfortunately, it seems to me that the authors are far less familiar with the problem of nucleation than they are with the methodological aspects of the simulations. Regarding their main claim, I think that a much simpler interpretation can be provided to explain the author's results. But more importantly, I think that by aiming at a detailed description, they miss the most significant bit of the heterogeneous nucleation problem, as discussed below. I enjoyed reading this manuscript and I think it contains very valuable state-of-the-art simulations. I hope it will find soon a suitable avenue for publication. But at the present stage with emphasis on the two states model that I don't trust I cannot recommend this for publication. A revised manuscript with a convincing response to point 1, a revised interpretation of the findings and some figures for the relative increase of nucleation rates relative to homogeneous nucleation would be very interesting.

Major

1. My first problem is related to the main claim: have the authors actually found a critical cluster? Of course, I do trust they have found an activated event. But perhaps not one that is sufficient to have the crystal grow boundless. From Figure 2, I find that all trajectories depicted flatten out after the first (or second) activation event. This certainly needs an explanation. So either those trajectories were not adequately chosen, or the crystal obtained in these simulations is still not critical. I have two thoughts on this issue. I) either the system has grown into a stable freezing layer, and further growth needs activation for whatever reason (this seems odd) or II) the wall-water energy cutoff that has been implemented at 1 nm suppresses further growth. This is not just a technicality. It would mean that growth is triggered by long range interactions from the wall. This is relevant, because the authors have chosen a fairly strong wall-water field, so the stabilization due to long range forces could well be spurious.

2. My second claim regards the choice of thermodynamic conditions. The authors are reporting mostly results that are significant at a temperature of 230 K. This is lower than the onset of homogeneous nucleation in pure undercooled water. Accordingly, the two nucleation mechanisms reported here are competing with yet another one, namely, homogeneous nucleation. So this finding, albeit interesting, appears not to be very significant, given that the 'non-classical' pathway reported here is not relevant at higher temperatures. This leads me to the formulation of what I believe is the most relevant question to answer in any study of heterogeneous nucleation: what is the relative increase with respect to homogeneous nucleation rates? If the methodology used here is good enough, this could be answered, since homogeneous nucleation rates for the TIP4P/Ice model have been reported in the literature.

3. Finally, I think that the interpretation given here of two different nucleation pathways is misleading. Of course, there are an infinite number of nucleation pathways, not just two, but it can often be useful to collect them into several different sets, if this provides some significant insight that could help understand the problem better. But I feel that this is not the case here. What I see from the very good figures chosen by the authors is that the rhombic ice does not get involved in the critical cluster. Instead, it appears to inhibit the nucleation of hexagonal ice. In a first stage, the substrate is covered randomly with patches of either hexagonal or rhombic ice. The latter does not promote further growth, because all hydrogen bonds are

saturated. Bulk ice can only grow on patches of hexagonal ice. The rhombic patches appear to be relatively stable sometimes, covering a substantial part of the substrate. In those cases, bulk ice can only grow on the remaining part of the substrate. So it has less space to grow, and must also grow in a vertical direction. Growth stops for a while until the rhombic ice is transformed into hexagonal ice and the nucleus can further spread in the lateral direction. So the interesting conclusion here is that at low temperatures rhombic and hexagonal ice compete to form a first layer. The latter inhibits further growth, the former fosters nucleation.

4. Related to this point, I think the explanation provided by the authors for the entropic role of rhombic ice is inaccurate. Entropy driven transformations are promoted at high temperature, while here it appears that the rhombic state becomes more likely at low temperatures. My bet is that calculation of the energy of one single monolayer of rhombic ice will be lower than that of hexagonal ice, the latter being entropically favored because of smaller topological constraints. I expect this is then the reason why rhombic ice disappears at higher temperature.

Other issues

1. I think the authors need to spend a paragraph either in the manuscript or in the supporting information explaining qualitatively what the MSM protocol is, without entering into the intricate details. Particularly I don't quite understand how simulations are being reported which span up to 4000 ns (Fig 2.a), while the Methods section indicates that only simulations of up to 1000 ns were undertaken. Are the paths in Fig.2a synthetic paths made from sticking different trajectories? This needs to be clarified for the interpretation of results. Could the authors observe spontaneous formation of their largest cluster at 230K or were subsequent simulations run from the largest cluster found in previous ones?

2. In page 9, the authors need to clarify when they talk about TS II whether it is the first or the second transition state along that path they refer to. Also clarify the details of the transition states in the broad sense. It had been mentioned that the final transition state along each path has equal size, but from figure 3.e transition states have significantly different sizes. This could be confusing. Notice that by taking rhombic ice away from TSII, the sizes become essentially the same, quite in agreement with the interpretation given in this report that the relevant cluster is actually that made of hexagonal ice. Also clarify whether the energy belonging to each cluster includes also water-wall energy. I think TSII has more energy because the cluster is taller. Due to the constraint that it cannot spread on rhombic ice, it is taller and therefore has larger line tension (which is proportional to the nucleus height).

3. I miss an analysis on the prevailing orientation of the hexagonal crystallite. Is it growing basal face up?

4. First sentence in page 11. Of course, free energies for ice nucleation rise as T increases. c.f. Classical Nucleation Theory.

5. Notice that a spherical cluster parameter is not relevant because simulations are performed well below the roughening temperature of ice. Therefore, one expects crystallites should be flat, rather than hemispherical.

6. It would be advisable to motivate the choice of the wall-water interaction with some figure of a real metal or crystal for comparison. By the way, I think never in the paper it is mentioned that it is nucleation from undercooled water that is studied here. It would also be of interest to mention whether the lateral cell dimensions chosen for the wurtzite substrate are multiples of the ice unit cell, as this will also have an impact on the final state within of a finite size system.

Reviewer #3 (Remarks to the Author):

The manuscript by Li et al is an interesting, well-performed, well-written piece of work concerning an ancient

and important problem that continues to puzzle the scientific community, i.e. nucleation. Specifically, here they study heterogeneous ice nucleation. Due to the somewhat rare event nature of the problem, the authors make use of markov state models and transition path theory. Several interesting and useful findings are reported which could in principle lead to controlled, rational design of crystallizing materials. The central finding in my view is that configurational entropy leads to non-classical pathway and surface energetics leads to classical pathway. Furthermore, at high temperatures the surface energetics become so weak that the entropy can not dominate and the classical pathway dominates entirely.

While I am happy to recommend publication as is, I have a few optional comments/questions:

1. Fig 1c is very instructive already, but perhaps more can be revealed from it. I believe no description is provided for what the slowest/second-slowest/third-slowest processes are.
2. Building on point 1, how does Fig 1c look at higher temp viz. 240 and 250K? The SI says same CVs were used to construct MSM at 230,240,250 K. However would the spectral-oASIS approach have given different number of CVs, and more importantly different gap between timescales of the 3 processes at different temperature, thereby predicting the key result of this paper? If no, why not? If yes, then why do MSM and TPT on top of spectral-oASIS?
3. The abstract says water-surface interactions present in heterogeneous ice nucleation complicate the process. I would argue that there are water-surface terms in homogeneous nucleation as well, i.e. between the nucleus and the surrounding water. Perhaps the authors could comment on this. Does this mean their arguments of entropy vs surface energetics could be extended to homogeneous nucleation as well?
4. It would be wonderful if the authors could directly upload somewhere the input 230,240,250 K trajectories used in this work as it would help with reproducibility as well as help others applying different methods to the problem.

- Pratyush Tiwary, University of Maryland

Response to reviewer #1's comments:

We thank the reviewer for recognizing of our methodological advance and computational efforts in studying heterogeneous ice nucleation by combining the Markov state models and the transition path theory. The reviewer also raised several comments, which are invaluable for us to improve our manuscript. In our revision, we have addressed the reviewer's major concerns by explaining the origin of the upper-bound in the kinetic trajectories in the original manuscript, and showing that ice crystals can grow boundlessly in our simulations with various cut-off values of the wall-water interactions. In addition, we demonstrate that the rate of heterogeneous ice nucleation is significantly faster than that of the homogeneous ice nucleation, which confirms the relevance of studying heterogenous ice nucleation in our system. We also emend the interpretation of the two nucleation pathways. Please see below for our point-to-point responses to all the comments:

1. My first problem is related to the main claim: have the authors actually found a critical cluster? Of course, I do trust they have found an activated event. But perhaps not one that is sufficient to have the crystal grow boundless. From Figure 2, I find that all trajectories depicted flatten out after the first (or second) activation event. This certainly needs an explanation. So either those trajectories were not adequately chosen, or the crystal obtained in these simulations is still not critical. I have two thoughts on this issue. I) either the system has grown into a stable freezing layer, and further growth needs activation for whatever reason (this seems odd) or II) the wall-water energy cutoff that has been implemented at 1 nm supresses further growth. This is not just a technicality. It would mean that growth is triggered by long range interactions from the wall. This is relevant, because the authors have chosen a fairly strong wall-water field, so the stabilization due to long range forces could well be supurious.

Reply: We appreciate this great comment from the reviewer. We want to clarify that the ice crystals in our molecular dynamics (MD) simulations can grow boundlessly. To demonstrate this point, we have extended four of our MD simulations that already formed ice and show that the number of hexagonal ice molecules can further increase from ~1200 (as shown in Fig. 2a-b of our original manuscript) to ~2500 for both 1-step (see Fig. R1) and 2-step (see Fig. R2) nucleation trajectories. Noticeably, when the number of hexagonal ice molecules reaches 2500, we can observe as many as 8 layers of ice on the surface (see representative snapshots in Fig. R1d-e and Fig. R2d-e). These observations clearly indicate that the reported ice nucleation processes in our study are truly activation events corresponding to the crossovers of free energy barriers, and can subsequently lead to the boundless growth of the ice crystals. Finally, we have also followed the reviewer's suggestion to examine whether the wall-water potential may hamper ice growth by varying the interaction cut-off values. Figure R3 illustrates that ice can grow boundlessly after nucleation at larger interaction cut-off values (e.g., 1.2 nm and 1.4 nm, as opposed to 1.0 nm), which indicates that the cut-off value in our original MD setting will not suppress ice formation.

We agree with the reviewer that the trajectories displayed in Fig. 2 of the original manuscript all seemingly flatten out when the number of hexagonal ice reaches ~1200. This is because those trajectories are not original MD simulation trajectories but are generated from the Markov State Model (MSM). In order to focus our analysis on the ice nucleation events, we set an upper-bound

(at 1200 hexagonal ice molecules) when building MSMs from MD simulation trajectories. As shown in Fig. R1-R2, the system has entered the stable ice growth stage at ~ 800 hexagonal ice molecules (e.g., trajectories in Fig. R1 & R2 display a linear increase in hexagonal ice when its number exceeds ~ 800), and we thus believe that the MSM with a cut-off at 1200 is well sufficient for us to analyze the ice nucleation events. From our MSM, we generated the trajectories displayed in the original Fig. 2 by performing the Markov Chain Monte Carlo (MCMC) sampling based on the transition probability matrix (see Fig. R4a and the italicized paragraphs below for details). As a result, these trajectories were intrinsically bound at ~ 1200 . We apologize for any confusion that Fig. 2a-b in our original manuscript may have caused.

To improve the clarity of our manuscript, we have updated Fig. 2a-b in the main text as Fig. R4b-c to only display MSM trajectories well before they reach the upper-bound, and have included the representative original MD trajectories (in Figs. R1 and R2) that display boundless ice growth in the SI as Figs. S5 and S6, respectively. In addition, we have included a more detailed description on how we perform MCMC sampling based on MSMs in the SI of the revised manuscript (see below). Again, we would like to thank this reviewer for his/her great comment that greatly helps us improve our manuscript.

***Maintext:** “...Representative kinetic trajectories corresponding to either one or two activation steps are displayed in Fig. 2a and 2b, respectively (see Methods and SI for more detail on generating these kinetic trajectories using Markov Chain Monte Carlo simulations based on the MSMs; representative original MD trajectories with one-step and two-step characteristics are also presented in Fig. S5 and S6, respectively) ...”*

***SI:** “... Markov State Models (MSMs) constructed from many simulations provide a promising approach to elucidate the kinetic ensemble of ice nucleation pathways. The basic concept of MSMs is to model the continuous dynamics as Markovian transitions among discrete partitions of configuration space. For ice nucleation, the MSMs first partition the configurational space of interest into a discrete number of states, and then compute the transition probability between pairs of the states at a discretized lag time τ from the parallel MD simulations. If the lag time τ is long enough to allow full relaxation within each state, the fast motions are integrated out and the model becomes Markovian, i.e., the probability for the system to visit a given state at time $t + \tau$ depends only on its current position at time t . Under this condition, the long-timescale dynamics can be modelled by a sequence of Markovian transitions among the discrete states based on the transition probability.”*

“...Note that when constructing MSMs, we chose a cut off value (in number of hexagonal ice molecules) when partitioning the configuration space sampled by the MD trajectories into a discrete set of micro-states. As shown in Fig. S5 and S6, after the number of hexagonal ice reaches ~ 800 , the system has already entered the stage of continuous ice growth, as indicated by the linear increase in the number of hexagonal ice. Thus, we believe that the MSM with a cut-off at 1200 hexagonal ice molecules is sufficient to analyze the ice nucleation events.”

“...with the transition probability matrix obtained from microstate MSMs, Markov Chain Monte Carlo simulations can be performed to simulate the Markov jumps based on the function of $p(t + \tau) = p(t)T(\tau)$ to correctly generate the long-time trajectories.”

Figure R1 (also SI Fig. S5 in the revised manuscript) Typical molecular dynamics trajectories of the classical one-step nucleation pathway at $T=230$ K. (a). Schematic of the free energy profile for the classical one-step ice nucleation pathway. (b)-(c). Evolutions of the number of hexagonal ice molecules in the largest ice nucleus, exhibiting one activation process. The gray regions are extended simulations showing that ice grows continuously after entering the growth stage. (d)-(e). Typical configurations of the largest ice nucleus for the trajectories in (b) and (c), respectively. The purple and green spheres denote the rhombic and hexagonal ice, respectively. The gray surfaces represent the substrate.

Figure R2 (also SI Fig. S6 in the revised manuscript) Typical molecular dynamics trajectories of the non-classical two-step nucleation pathway at $T=230$ K. (a). Schematic of the free energy profile for the non-classical two-step ice nucleation pathway. (b)-(c). Evolutions of the number of hexagonal ice molecules in the largest ice nucleus, exhibiting two activation processes. The gray regions are extended simulations showing that ice grows continuously after entering the growth stage. (d)-(e). Typical configurations of the largest ice nucleus for the trajectories in (b) and (c), respectively. The purple and green spheres denote the rhombic and hexagonal ice, respectively. The gray surfaces represent the substrate.

Figure R3 Typical MD trajectories conducted with larger wall-water interaction cut-off distances (i.e., 1.2 nm and 1.4nm), showing that ice can grow boundlessly after nucleation and the cut-off distance will not induce the upper bound. (a)–(b) Evolutions of the number of hexagonal ice molecules in the largest ice nucleus when the cut-off distance for wall-water interactions is 1.2 nm. (c)–(d) Evolutions of the number of hexagonal ice molecules in the largest ice nucleus when the cut-off distance for wall-water interactions is 1.4 nm.

Figure R4 (panel b-c are also Fig. 2a-b in the revised manuscript) (a). Transition probability matrix with $N=1000$ microstates, the element P_{ij} in the matrix represents the transition probability from microstate i to j . (b). Representative kinetic trajectories with one activation crossing over one free energy barrier. (c). Representative kinetic trajectories with more than one activation crossing over two or more free energy barriers.

2. My second claim regards the choice of thermodynamic conditions. The authors are reporting mostly results that are significant at a temperature of 230 K. This is lower than the onset of homogeneous nucleation in pure undercooled water. Accordingly, the two nucleation mechanisms reported here are competing with yet another one, namely, homogeneous nucleation. So this finding, albeit interesting, appears not to be very significant, given that the ‘non-classical’ pathway reported here is not relevant at higher temperatures. This leads me to the formulation of what I believe is the most relevant question to answer in any study of heterogeneous nucleation: what is the relative increase with respect to homogeneous nucleation rates? If the methodology used here is good enough, this could be answered, since homogeneous nucleation rates for the TIP4P/Ice model have been reported in the literature.

Reply: We would like to thank the reviewer for his/her good comment. We have followed the reviewer’s suggestion to compare homogeneous and heterogeneous ice nucleation rates. For our system (with the TIP4P/ice water model) at $T = 230$ K, we estimated a heterogeneous ice nucleation rate of $J_{\text{het}} = 3.5 \times 10^{30 \pm 1} \text{ s}^{-1} \text{ m}^{-3}$ ($J_{\text{het}} = 1/(t_{\text{MFPT}}V)$, where t_{MFPT} is the mean first passage time for the system to reach the stable growth stage and V is the volume of water in the system). We note that a previous study also reported a fast heterogeneous ice nucleation rate of $J_{\text{het}} = 10^{26 \pm 2} \text{ s}^{-1} \text{ m}^{-3}$ on another surface (i.e., Kaolinite surface) using the TIP4P/ice model at 230K (*J. Phys. Chem. Lett.* 7, 2350–2355 (2016)). These estimated heterogeneous ice nucleation rates are orders of magnitude faster than that for the homogeneous ice nucleation: $J_{\text{hom}} = 10^{5.9299 \pm 0.6538} \text{ s}^{-1} \text{ m}^{-3}$ for the same water model at 230 K (*PNAS* 112, 10582-10588, (2015)). These observations further confirm the relevance of studying heterogenous ice nucleation as a major mode in competition with homogeneous ice nucleation at $T=230$ K. We also note that two recent experimental studies have successfully prepared supercooled water with temperatures as low as 230.6K (*Phys. Rev. Lett.* 120, 015501 (2018)) and 227.7K (*Science* 358, 1589, (2017)), respectively. These studies suggest that the investigation of heterogenous ice nucleation at $T=230$ K (-42.2 °C for TIP4P/ice water) can still offer useful insights into the nucleation process.

To include these points in our revised manuscript, we have inserted the following sentences in the revised manuscript. We would like to thank the reviewer again for these instructive suggestions.

Maintext:

“... our MSMs show that when water is supercooled to 230 K, ice nucleation...”

“...More importantly, the heterogeneous ice nucleation rate can be estimated from the MFPTs as $J_{\text{het}} = 3.5 \times 10^{30 \pm 1} \text{ s}^{-1} \text{ m}^{-3}$ for our system at $T=230$ K (see Sec. 5 in SI for the method), which is orders of magnitude faster than the rate for homogeneous ice nucleation ($J_{\text{hom}} = 10^{5.9299 \pm 0.6538} \text{ s}^{-1} \text{ m}^{-3}$)⁵⁶. We also notice that a previous study⁵⁹ reported a fast heterogeneous ice nucleation rate of $J_{\text{het}} = 10^{26 \pm 2} \text{ s}^{-1} \text{ m}^{-3}$ on a Kaolinite surface using the same water model at 230K. These underline the relevance of studying heterogenous ice nucleation as the major mode in competition with homogeneous ice nucleation when water is supercooled to $T=230$ K.”

SI: the method to estimate the ice nucleation rate is added:

“Subsequently, the nucleation rate J can be estimated by $J = \frac{1}{t_{MFPT}V}$, where t_{MFPT} is the MFPT for the system to enter the growth stage, and V is the volume of water in the system^{23,24}. Specifically, $t_{MFPT} = \frac{\sum_i p_i t_{MFPT}^i}{\sum_i p_i}$, where p_i is the percentage of flux i , and t_{MFPT}^i is the MFPT for the system from macro-

state I to macro-states IV or V for flux i . Note that the heterogeneous nucleation rate should normally be evaluated using the surface area (for nucleation on a surface). Here, a volume-based nucleation rate is used to allow for a direct comparison with homogeneous nucleation²³.”

3. Finally, I think that the interpretation given here of two different nucleation pathways is misleading. Of course, there are an infinite number of nucleation pathways, not just two, but it can often be useful to collect them into several different sets, if this provides some significant insight that could help understand the problem better. But I feel that this is not the case here. What I see from the very good figures chosen by the authors is that the rhombic ice does not get involved in the critical cluster. Instead, it appears to inhibit the nucleation of hexagonal ice. In a first stage, the substrate is covered randomly with patches of either hexagonal or rhombic ice. The latter does not promote further growth, because all hydrogen bonds are saturated. Bulk ice can only grow on patches of hexagonal ice. The rhombic patches appear to be relatively stable sometimes, covering a substantial part of the substrate. In those cases, bulk ice can only grow on the remaining part of the substrate. So it has less space to grow, and must also grow in a vertical direction. Growth stops for a while until the rhombic ice is transformed into hexagonal ice and the nucleus can further spread in the lateral direction. So the interesting conclusion here is that at low temperatures rhombic and hexagonal ice compete to form a first layer. The latter inhibits further growth, the former fosters nucleation.

Reply: We would like to thank the reviewer for this great comment. We appreciate the nice explanation from the reviewer regarding the competitive formation of rhombic and hexagonal ice during the early stage of ice nucleation. In fact, the mechanism proposed by the reviewer is not mutually exclusive from ours. Two key points to reconcile these two mechanisms are: (a) the faster rate to form rhombic patches than hexagonal patches on our surface; and (b) the relatively rapid conversion of rhombic ice into hexagonal ice. For the first point, we show that the formation of the ice nucleus containing a mixture of rhombic ice patches and hexagonal ice patches (i.e., corresponding to a MFPT of 1.34 μ s for the crossover of the 1st free energy barrier in our non-classical nucleation pathway, see Fig. R5a) is $\sim 20\%$ faster than the formation of the ice nucleus containing purely hexagonal ice patches on the surface (i.e. corresponding to a MFPT of 1.65 μ s for the crossover of the free energy barrier on the classical pathway, see Fig. R5a). This faster rate along the first step of the non-classical pathway is attributed to the configurational entropy from the mixing of rhombic and hexagonal ice, leading to a faster growth of the ice mixture (overcoming the first free energy barrier in the non-classical pathway, see Fig. R5a). For the second point, we show that the conversion of the rhombic ice into hexagonal ice is not the rate-limiting step (i.e., overcoming the second free energy barrier in the non-classical pathway). For example, the MFPTs

for such conversion via transitions of III→IV and III→V occur at around 0.59 and 1.17 μs, respectively. As a result, we observe the co-existence of the two pathways in our system even though the classical nucleation pathways indeed have a higher flux (classical vs. non-classical: 61.9% vs. 38.1%).

In our revised manuscript, we reconciled our original mechanism with the reviewer's proposed one, and refined our interpretations of the nucleation mechanisms. The hexagonal ice and rhombic ice compete to form at the early stage, which leads to the bifurcation of the flux into two distinguished pathways, i.e., the classical one-step and the non-classical two-step nucleation pathways. Specifically, the direct increase in hexagonal ice with little involvement of rhombic ice describes the classical one-step nucleation pathway; in contrast, the significant increase of rhombic ice mixed with hexagonal ice corresponds to the first step of the nonclassical two-step pathway (i.e., the crossover of the 1st free energy barrier along the nonclassical pathway). The conversion of the rhombic ice patches back to hexagonal ice, which is relatively fast, corresponds to the second step of the nonclassical two-step pathway (i.e., the crossover of the 2nd free energy barrier). After this step, subsequent ice growth occurs. In this sense, the formation of the ice nucleus mixed with rhombic and hexagonal ice serves as an intermediate that gives rise to the non-classical two-step nucleation pathway.

We would like to thank the reviewer again for the nice interpretation. We acknowledge that we did not explicitly explain the difference between the classical and non-classical nucleation pathways in our original manuscript. In the revised manuscript, we have emended the interpretation and incorporated the descriptions of the two pathways (see below). In addition, we have updated Fig. 3 using Fig. R5.

“... Figure 2c shows that the rhombic ice and hexagonal ice compete to form during the early stage of nucleation, causing the flux to bifurcate into two distinguished pathways. Specifically, the flux that transitions from macro-states I→II→IV (grey arrow), suggests a direct formation of hexagonal ice with little involvement of rhombic ice, which corresponds to the classical one-step nucleation pathway. In the other direction, the fluxes that transition from macro-states I→II→III (purple, orange, and red arrows) suggest a significant increase in rhombic ice and hexagonal ice at the first step. The rhombic ice then transforms back into hexagonal ice via macro-states III→IV/V as the second step, as shown by the structural evolutions of the nucleus along the three fluxes (i.e., purple, orange, and red). These fluxes (including fluxes in purple, red, and orange) belong to the non-classical two-step nucleation pathway^{11,25,103} since they all involve the intermediate development of characteristic pre-nucleation structures, i.e., a mixture of rhombic and hexagonal ice, which is different from the classical description of the direct formation of hexagonal ice. After the conversion of rhombic ice into hexagonal ice, the fluxes coalesce and enter the growth stage, as shown by the further growth of hexagonal ice...”

“...Interestingly, the MFPTs for the transition from macro-states I to III is shorter than that from macro-states I to IV (i.e., 1.34 vs. 1.65 μs). This suggests that the 1st free energy barrier along the non-classical pathway is smaller than that for the classical pathway.”

“...after which the rhombic ice is further converted into hexagonal ice via transitioning from state III→IV or III→V (at around 0.59 and 1.17 μs, respectively)”

Figure R5 (also Fig. 3 in the revised manuscript). Critical nuclei along the two nucleation pathways. The critical nuclei characterized by the transition states (TSs) are obtained by two independent transition path theory (TPT) analyses, in which the source is localized in macro-state I, where the separation of the two paths initiates, and the sinks are selected independently at macro-states III and IV, where the separation is finished. (a) Schematic of the free energy landscape for the formation of the ice nucleus proceeding via two separate pathways. Macro-states I to IV (gray) represents the crossover of the free energy barrier along the classical ice nucleation pathway with direct formation of hexagonal ice. Macro-states I to III (orange) represents the crossover of the 1st free energy barrier along the non-classical two-step ice nucleation pathway, where the ice nucleus is comprised of a mixture of rhombic and hexagonal ice. (b) Representative configurations (top: front view; bottom: [101] view) of the critical nucleus for TS II. (c) Representative configurations (top: front view; bottom: [101] view) of the critical nucleus for TS I. (d) Comparison of the two TSs with respect to the average potential energy per water molecule of the critical nucleus, where the error bars represent the standard errors with samples obtained by bootstrapping. (e) Comparison of the two TSs with respect to the amount of rhombic and hexagonal ice in the critical nucleus. (f) Comparison of the two TSs with respect to the disorderliness (defined as the ratio of number of rhombic and hexagonal ice) of the nucleus.

4. Related to this point, I think the explanation provided by the authors for the entropic role of rhombic ice is inaccurate. Entropy driven transformations are promoted at high temperature, while here it appears that the rhombic state becomes more likely at low temperatures. My bet is that calculation of the energy of one single monolayer of rhombic ice will be lower than that of hexagonal ice, the latter being entropically favored because of smaller topological constraints. I expect this is then the reason why rhombic ice disappears at higher temperature.

Reply: We would like to thank the reviewer for this great comment. As the reviewer pointed out, the entropy driven transformations in general become more prominent at higher temperatures. However, at low temperatures, the presence of rhombic ice in the critical nucleus in our system (transition state of the 1st step in the non-classical nucleation pathway, Fig. R5a) plays a unique role by inducing the conformational disorder of the critical nucleus via the mixing of rhombic and hexagonal ice. This increases the number of configurations (i.e., ways to mix two kinds of ice molecules) and thus enhances the configurational entropy of the transition state. As a result, the favorable entropy term $-TS$ will stabilize the transition state and lower the activation free energy for the non-classical nucleation pathway. In terms of energy, we show that the transition state along the classical nucleation pathway (consisted of purely hexagonal ice) actually has a more favorable potential energy than that of the 1st step of the non-classical pathway (see Fig. R6a). It is worth noting that the average potential energy for the rhombic ice at the interface is also slightly higher than that for the 1st layer of hexagonal ice (see Fig. R6b), and rhombic ice also has a slightly weaker wall-ice interaction energy as shown in Fig. R6c. Despite these slightly unfavorable energetic terms, the favorable configurational entropic contributions will still render the transition rates for the 1st step of the non-classical pathway to be slightly faster than the classical pathway, as indicated by its shorter transition time (MFPT of 1.34 μ s vs. 1.65 μ s, see Fig. R5a).

At high temperatures, the total entropy of the system indeed increases and thus tends to drive the system into liquid water, leading to an increase in the free energy barrier for the formation of ice along both pathways simultaneously. The increase in the free energy barrier will correspond to a larger critical nucleus. For example, in our system, the size of the critical nucleus (analyzed by the transition path theory) increases with temperature for the classical pathway (i.e., the total number of ice molecules = 475, 593, and 665 for $T = 230, 240,$ and 250 K, respectively), which generally agrees with classical nucleation theory and previous work (*Phys. Rev. Lett.* 122, 245501 (2019)). As a result, the potential energy difference for the two transition states along the two pathways will increase with nucleus size. However, in terms of configurational entropy, we anticipate that the difference between the two pathways may not increase much since rhombic ice mainly forms on the first layer, and the increase in critical nucleus size (i.e., from the formation of mainly hexagonal ice after the first layer is occupied) will not increase the configurational entropy enough to compensate for the potential energy difference. As a result, the potential energy difference overrides the configurational entropy compensation, making the classical nucleation pathway more accessible than the non-classical pathway at high temperatures.

To include the above discussions, we have modified the explanation of the temperature effects on the two pathways in the revised manuscript as below:

Abstract: “Furthermore, we discover that, at elevated temperatures, the nucleation process prefers to proceed via the classical pathway, as opposed to the non-classical pathway, since the potential energy contributions override the configurational entropy compensation”.

Maintext: “...by elevating the temperature, we discover that the nucleation process shifts significantly towards the classical pathway, mainly because the potential energy difference, which favors the classical pathway, prevails over the configurational entropy compensation.”

“...At higher T , the total entropy of the system increases and tends to drive the system into liquid water, leading to an increase in the free energy barrier along both pathways simultaneously. The elevated free energy barrier will correspond to a larger critical nucleus. For example, in our system, the size of the critical nucleus for the classical pathway also increases with temperature (i.e., the total number of ice molecules = 475, 593, and 665 when $T = 230, 240,$ and 250 K, respectively), which agrees with CNT and previous work³⁰. As a result, the potential energy difference between the two TSs along the two pathways will increase with nucleus size. In terms of configurational entropy, we anticipate that the difference between the two pathways may not increase much since the rhombic ice mainly forms in the first layer, and the increase in critical nucleus size (i.e., from the formation of mainly hexagonal ice since the first layer has been fully occupied) will not increase the configurational entropy enough to compensate for the potential energy difference. Therefore, the potential energy difference overrides the configurational entropy compensation, making the classical nucleation pathway more accessible than the non-classical pathway.”

Figure R6 (Panel a is also plotted in Fig. 3d; Panel b and c are also plotted in Fig. S8 in the SI of the revised manuscript) (a) Average total potential energy of transition states I (i.e., state of the ice nucleus at the free energy barrier for the classical one-step pathway) and II (i.e., state of the mixed ice nucleus at the 1st free energy barrier for the non-classical two-step pathway). (b). Average total potential energy for the 1st-layer rhombic and 1st-layer hexagonal ice in contact with the surface. (c). Average ice-wall interaction energy for the 1st-layer of rhombic and 1st-layer of hexagonal ice in contact with the surface. The hexagonal ice has a stronger interaction with the surface than the rhombic ice.

Other issues

5. I think the authors need to spend a paragraph either in the manuscript or in the supporting information explaining qualitatively what the MSM protocol is, without entering into the intricate details. Particularly I don't quite understand how simulations are being reported which span up to 4000 ns (Fig 2.a), while the Methods section indicates that only simulations of up to 1000 ns were undertaken. Are the paths in Fig.2a synthetic paths made from sticking different trajectories? This needs to be clarified for the interpretation of results. Could the authors observe spontaneous formation of their largest cluster at 230K or where subsequent simulations run from the largest cluster found in previous ones?

Reply: We thank the reviewer for this good comment. We have followed the reviewer's suggestion to add a general explanation of Markov state model (MSM) (see below). As discussed in the response to this reviewer's comment #1, we indeed observe the spontaneous formation of the largest ice nucleus at $T=230\text{K}$ as shown by the typical original MD trajectories in Fig. R1 and R2. As pointed out by the reviewer, the long kinetic trajectories (paths shown in original Fig. 2a) are indeed not the original MD simulation trajectories, but are synthetic trajectories generated from the MSM (MSMs are constructed based on many MD trajectories at shorter length).

Again, we would like to thank the reviewer for his/her good suggestion to provide a general explanation of Markov state models to improve the manuscript. In the revised manuscript, we have clarified the method we used to generate the kinetic trajectories and included the discussion of MSM in the SI.

Maintext: "... Representative kinetic trajectories corresponding to either one or two activation steps are displayed in Fig. 2a and 2b, respectively (see Methods and SI for more detail on generating these kinetic trajectories using Markov Chain Monte Carlo simulations based on the MSMs; representative original MD trajectories with one-step and two-step characteristics are also presented in Fig. S5 and S6, respectively) ..."

SI: "... Markov State Models (MSMs) constructed from many simulations provide a promising approach to elucidate the kinetic ensemble of ice nucleation pathways. The basic concept of MSMs is to model the continuous dynamics as Markovian transitions among discrete partitions of configuration space. For ice nucleation, the MSMs first partition the configurational space of interest into a discrete number of states, and then compute the transition probability between pairs of the states at a discretized lag time τ from the parallel MD simulations. If the lag time τ is long enough to allow full relaxation within each state, the fast motions are integrated out and the model becomes Markovian, i.e., the probability for the system to visit a given state at time $t + \tau$ depends only on its current position at time t . Under this condition, the long-timescale dynamics can be modelled by a sequence of Markovian transitions among the discrete states based on the transition probability..."

6. In page 9, the authors need to clarify when they talk about TS II whether it is the first or the second transition state along that path they refer to. Also clarify the details of the transition states in the broad sense. It had been mentioned that the final transition state along each path has equal size, but from figure 3.e transitions states have significantly different sizes. This could be confusing. Notice that by taking rhombic ice away from TSII, the sizes

become essentially the same, quite in agreement with the interpretation given in this report that the relevant cluster is actually that made of hexagonal ice. Also clarify whether the energy belonging to each cluster includes also water-wall energy. I think TSII has more energy because the cluster is taller. Due to the constraint that it cannot spread on rhombic ice, it is taller and therefore has larger line tension (which is proportional to the nucleus height).

Reply: We would like to thank the reviewer for these good suggestions. We totally agree with the reviewer that the definitions of transition states and macro-states need to be clarified. The transition state (TS) is located at the highest free energy point along a reaction coordinate that can describe the ice nucleation (i.e., free energy barriers), and the TS configurations should then correspond to the critical ice nucleus. In contrast, the macro-states obtained in MSMs represent metastable regions (i.e., free energy basins). In this sense, macro-states represent the states with low free energy and are separated by free energy barriers (corresponding to the TSs), as illustrated in Fig. R5a.

In the original manuscript, we did not state that the transition state along each path has equal size. Instead, we have shown that macro-states VII and VIII can be combined as the final state as they are consisted of configurations containing similar number of ice molecules. As explained in the previous paragraph, these macro-states (VII and VIII, as shown in Fig. R7) are not transition states but are metastable states. In Fig. 3e, we identified the transition state using the transition path theory, and these transition states (TS I and TS II) indeed contain configurations with various sizes of ice molecules.

In terms of energy, the total potential energy of the ice nucleus for the TS includes the potential energy within the ice nucleus, ice-wall interaction energy, and ice-liquid interaction energy. Indeed, the longer shape of the nucleus (TS II) will have more water at the ice-liquid interface, which will contribute to a higher total potential energy on average. We also note that the rhombic ice has higher average potential energy than hexagonal ice, as shown in Fig. R5b, which will also contribute to the higher total potential energy of TS II (first step of the non-classical pathway) than TS I (classical pathway).

In the revised manuscript, we have clarified the definitions of transition states and macro-states. We also defined the total potential energy and explained the potential source that leads to the potential energy difference in the two transition states.

“... The macro-states from MSMs represent metastable regions (i.e., free energy basins), and are separated by free energy barriers (i.e., transition states) ...”

“... The free energy landscape where the nucleation pathway bifurcates with the transition states (TSs) at the saddle points which govern the transitions, is illustrated in Fig.3a. Specifically, TS I in Fig. 3c corresponds to the critical ice nucleus for the classical one-step pathway; and TS II in Fig. 3b corresponds to the critical ice nucleus for the 1st barrier of the non-classical two-step pathway (see Methods for the identification of the TSs using TPT).”

“... However, the difference in average potential energy (including the potential energy within the ice nucleus, ice-surface interaction energy, and ice-liquid water interaction energy)...”

“... This can be partially attributed to the surface, which has stronger interactions with the first-layer of hexagonal ice than rhombic ice, as shown in Fig. S8. Another factor that might lead to the lower potential energy for TS I can be the noticeably smaller amount of ice at the liquid-ice interface compared to TS II (taller ice nucleus structures in TS II than those in TS I, see Fig. 3b and 3c), as ice at the liquid-ice interface is generally less stable with higher potential energy¹⁰⁴.”

Figure R7 (also Fig. 2c in the manuscript) The kinetic pathways of HIN: coexistence of the classical and non-classical nucleation pathways. For each macro-state, a representative front view of the ice configuration is shown, where the hexagonal and rhombic ice molecules are represented in green and purple spheres, respectively. The macro-states VII and VIII are categorized as a final state, around or after which the system enters the period of ice growth.

7. I miss an analysis on the prevailing orientation of the hexagonal crystallite. Is it growing basal face up?

Reply: We would like to thank the reviewer for his/her question. The hexagonal ice grows on its first prism plane, as shown by the different views in Fig. R8. To include this in our revised manuscript, we have added the following statement in the SI:

“... The hexagonal ice nucleates and then grows mainly on its first prism plane...”

Figure R8 Various snapshot of the configurations of hexagonal ice on the surface. The green spheres and gray surface denote hexagonal ice and the surface, respectively.

8. First sentence in page 11. Of course, free energies for ice nucleation rise as T increases. C.f. Classical Nucleation Theory.

Reply: We would like to thank the reviewer for his/her comment. Indeed, based on classical nucleation theory, the free energy barrier for ice nucleation increases as T increases. We revised the sentence in the main text accordingly.

“This qualitatively agrees with CNT and previous studies^{30,105}, which showed that the free energy barrier for ice nucleation rises as T increases.”

9. Notice that a spherical cluster parameter is not relevant because simulations are performed well below the roughening temperature of ice. Therefore, one expect crystallites should be flat, rather than hemispherical.

Reply: We thank the reviewer for this good comment. We fully agree with the reviewer that the ice nucleus is relatively flat as opposed to spherical for heterogeneous ice nucleation in our system. Consequently, the spherical parameter is not selected as the collective variable to construct the Markov state models.

To include this in the revised manuscript, we have modified the discussion as follows:

“Interestingly, the spherical parameter was not selected, most likely due to the ice nucleus being relatively flat. This implies that the spherical parameter cannot properly describe the kinetics of HIN in our system.”

10. It would be advisable to motivate the choice of the wall-water interaction with some figure of a real metal or crystal for comparison. By the way, I think never in the paper it is mentioned that it is nucleation from undercooled water that is studied here. It would also be of interest to mention whether the lateral cell dimensions chosen for the wurtzite substrate are multiples of the ice unit cell, as this will also have an impact on the final state within of a finite size system.

Reply: We would like to thank the reviewer for these good suggestions. The wall-water interaction strength ($\epsilon_{ws}=4.98829$ kJ/mol) is close to, but slightly less than, that of gold-water interactions. (i.e., water-gold interaction: $\epsilon_{O-Au}=5.2362576$ kJ/mol in *Appl. Phys. Lett.* 103, 253104 (2013); water-copper interaction: $\epsilon_{O-Cu}=6.1625166$ kJ/mol in *Applied Thermal Engineering* 62, 607-612 (2014)). Indeed, the temperature of water is supercooled, and the lateral cell dimensions chosen for the wurtzite substrate match the hexagonal ice unit cell to minimize potential strain energy induced by lattice mismatch.

To include these points in our revised manuscript, we have added the followings:

“...our MSMs show that when water is supercooled to 230 K, ice nucleation...”

“These underline the relevance of studying heterogenous ice nucleation as the major mode in competition with homogeneous ice nucleation when water is supercooled to $T=230$ K.”

“... and the unit cell parameters were chosen to match the hexagonal ice to minimize strain energy due to lattice mismatch”

SI:

“The water-surface interaction strength is close to, but a bit less than, that of gold-water interactions, i.e., $\epsilon_{O-Au}=5.2362576$ kJ/mol⁸.”

Response to reviewer #3's comments:

We appreciate the reviewer's recognition of our work's central findings regarding the competitive classical and non-classical pathways and the underlying mechanisms that leads to the two pathways. We would also like to thank the reviewer for his acknowledgement of our methodological advance in studying ice nucleation. He also raised a few comments/questions to further improve our manuscript, which are very helpful and instructive. Please see below for our point-to-point response to these comments:

1. Fig 1c is very instructive already, but perhaps more can be revealed from it. I believe no description is provided for what the slowest/second-slowest/third-slowest processes are.

Reply: We thank the reviewer for this good comment. We have followed the reviewer's suggestion to elucidate the contributions of five chosen collective variables (CVs) to the three slowest timescales (see Fig. R9) via their relative contributions to three leading eigenvectors from our spectral-oASIS analysis, as shown in Fig. 1c. Interestingly, we find that the three slowest timescale processes are strongly correlated with the formation of ice in different layers, and are also consistent with our proposed ice nucleation mechanisms. For instance, as shown in Fig. R9 (top panel), the slowest timescale is mainly related to the CVs that represent the number of rhombic ice molecules in the largest ice nucleus and the total number of ice molecules. These two CVs can well describe the formation of ice in the first layer in contact with the surface, as rhombic ice mainly forms in the first layer and the number of hexagonal ice molecules can be obtained by deducting the number of rhombic ice molecules from the number of total ice molecules. Consistently, the formation of ice patches in the first layer indeed corresponds to the rate limiting step (the slowest timescale process) in both of our classical (formation of purely hexagonal ice mainly in the first layer) and non-classical ice nucleation pathways (formation of the mixture of rhombic ice and hexagonal ice mainly in the first layer). The second-slowest timescale and third-slowest timescale are mainly related to the formation of hexagonal ice in the third layer and second layer of the largest ice nucleus, respectively. Notably, the timescale for the formation of hexagonal ice in the third layer (middle panel, Fig. R9) is found to be slower than that in the second layer (bottom panel, Fig. R9). This observation actually underlines the second step in our two-step non-classical nucleation pathway, where the rhombic ice needs to be converted into hexagonal ice. This conversion process mainly involves the first two layers of ice, as the transformation of the rhombic ice molecule in the first layer will free one of its -OH bonds to quickly form a hydrogen bond with a hexagonal ice molecule in the 2nd layer. As a result, the growth of the 3rd layer of hexagonal ice is contingent upon the completion of this conversion process, rendering it to be the second-slowest timescale.

To include the above discussion in our revised manuscript, we have added Fig. R9 as Fig. S13 and incorporated an explanation in the methods section under "selections of representative collective variables" in the SI. Again, we would like to thank the reviewer for this insightful comment.

Maintext: *"The slowest timescales are strongly correlated with ice formation in different layers, as discussed in Sec. 3.1 in the SI."*

SI: “... Furthermore, we have elucidated the contributions of five chosen collective variables (CVs) to the three slowest timescales (see Fig. S13) via their relative contributions to three leading eigenvectors from our spectral-oASIS analysis, as shown in Fig. 1c. Interestingly, we find that the three slowest timescales are strongly correlated with the formation of ice in different layers. For instance, as shown in Fig. S13 (top panel), the slowest timescale is mainly related to the CVs that represent the number of rhombic ice molecules in the largest ice nucleus and the total number of ice molecules. These two CVs can well describe the formation of ice in the first layer in contact with the surface, as rhombic ice mainly forms in the first layer and the number of hexagonal ice molecules can be obtained by deducting the number of rhombic ice molecules from the number of total ice molecules. Consistently, the formation of ice patches in the first layer indeed corresponds to the rate limiting step (the slowest timescale process) in both of our classical (formation of purely hexagonal ice mainly in the first layer) and non-classical ice nucleation pathways (formation of the mixture of rhombic ice and hexagonal ice mainly in the first layer). The second-slowest timescale and third-slowest timescale are mainly related to the formation of hexagonal ice in the third layer and second layer of the largest ice nucleus, respectively. Notably, the timescale for the formation of hexagonal ice in the third layer (middle panel, Fig. S13) is found to be slower than that in the second layer (bottom panel, Fig. S13). This observation underlines the second step in our two-step non-classical nucleation pathway, where the rhombic ice needs to be converted into hexagonal ice. This conversion process mainly involves the first two layers of ice, as the transformation of the rhombic ice molecule in the first layer will free one of its -OH bonds to quickly form a hydrogen bond with a hexagonal ice molecule in the 2nd layer. As a result, the growth of the 3rd layer of hexagonal ice is contingent upon the completion of this conversion process, rendering it to be the second-slowest timescale...”

Figure R9 (also Fig. S13 in the SI of the revised manuscript) Contributions of the CVs to the timescales. To evaluate the contributions, we took the three leading eigenvectors that correspond to the three slowest timescales, and then normalized them based on the squared sum of their components. The normalized

square of each component in each eigenvector, V^2 , is plotted as the relative contribution. A higher V^2 value indicates a stronger contribution to the slowest timescales. The leading eigenvectors were obtained from the time-lagged correlation matrix from the Spectral-oASIS analysis with the five chosen CVs as shown in Fig. 1c.

2. Building on point 1, how does Fig 1c look at higher temp viz. 240 and 250K? The SI says same CVs were used to construct MSM at 230, 240, 250 K. However, would the spectral-oASIS approach have given different number of CVs, and more importantly different gap between timescales of the 3 processes at different temperature, thereby predicting the key result of this paper? If no, why not? If yes, then why do MSM and TPT on top of spectral-oASIS?

Reply: We would like to thank the reviewer for this good comment. We have followed the reviewer's suggestion to conduct spectral-oASIS analysis at $T=240$ K and 250 K. As shown in Fig. R10, we also need as many as around five CVs in order to reproduce the slowest timescales of the completed sets of CVs. Furthermore, we found that the composition of the selected CVs largely overlapped with those selected at 230K (In both $T=240$ K and 250 K, the five chosen CVs by Spectral-oASIS are: the number of ice molecules in the largest ice nucleus, number of hexagonal ice in the 2nd, 3rd, and upper layers of the largest ice nucleus, and the number of ice molecules in the largest hexagonal ice nucleus). The most notable difference is that the CV that represents the number of rhombic ice (one of the two CVs corresponding to the slowest timescale at 230K, see top panel of Fig. R9a) is no longer selected at 240K and 250K. This observation agrees well with one of our key conclusions that the non-classical two-step pathway (where the rate-limiting step involves the formation of rhombic ice) becomes less significant as temperature increases. Even though the Spectral-oASIS analysis can nicely select CVs to reflect the change in slowest timescales at different temperatures, it couldn't explicitly elucidate two parallel pathways of ice nucleation and their flux (e.g., the number of rhombic ice and hexagonal ice molecules in the first layer correspond to the rate-limiting step of both classical and non-classical pathways at 230K). Therefore, we believe that it will still be useful to construct MSM and perform TPT analysis using these selected CVs to facilitate the comprehension of the ice nucleation mechanisms. In order to compare the flux of the classical and non-classical pathways at different temperatures, we adopted the same CVs and state-definition obtained at $T=230$ K across all three temperatures. Finally, we performed a control analysis to show that these five CVs chosen at 230K (largely overlapped with CVs chosen by Spectral-oASIS at 240K and 250K) can still reasonably reproduce the slowest timescales at these two higher temperatures (see right panels in Fig. R10a and b for 240K and 250K, respectively).

Again, we would like to thank the reviewer for this valuable comment. In the revised manuscript, we have added the discussion above in the methods section in the SI under construction and validation of MSMs at higher temperatures. In addition, we have included Fig. R10 in the revised manuscript as Fig. S15.

"... Before the MSM construction, we have also conducted spectral-oASIS analysis at $T=240$ K and 250 K. As shown in Fig. S15, we also need around five CVs to reproduce the slowest timescales of the completed sets of CVs. Furthermore, we found that the selected CVs is largely overlapped with those selected at 230K. For instance, in both $T=240$ K and 250 K, the five chosen CVs by Spectral-oASIS are: the number of ice

molecules in the largest ice nucleus, number of hexagonal ice in the 2nd, 3rd, and upper layers of the largest ice nucleus, and the number of ice molecules in the largest hexagonal ice nucleus. The most notable difference is that the CV that represents the number of rhombic ice is no longer selected at 240K and 250K. This observation agrees well with one of our key conclusions that the non-classical two-step pathway (where the rate-limiting step involves the formation of rhombic ice) becomes less significant as temperature increases. In order to compare the flux of the classical and non-classical pathways at different temperatures, we adopted the same CVs and state-definition obtained at $T=230\text{K}$ across all three temperatures. In addition, we have also performed a control analysis to show that these five CVs chosen at 230K (largely overlapped with CVs chosen by Spectral-oASIS at 240K and 250K) can still reasonably reproduce the slowest timescales at these two higher temperatures (see right panels in Fig. S15a and b for 240K and 250K, respectively).”

Figure R10 (also Fig. S15 in the SI of the revised manuscript) Timescales for the dynamics of the system with various numbers of CVs for $T=240\text{K}$ (a) and 250K (b), respectively. The left panels present timescales analysed by spectral-oASIS and the right panels are the timescales when using the CVs obtained in $T=230\text{K}$.

3. The abstract says water-surface interactions present in heterogeneous ice nucleation complicate the process. I would argue that there are water-surface terms in homogeneous nucleation as well, i.e. between the nucleus and the surrounding water. Perhaps the authors could comment on this. Does this mean their arguments of entropy vs surface energetics could be extended to homogeneous nucleation as well?

Reply: We would like to thank reviewer for this insightful comment. The water-surface interactions we mentioned in the abstract denote the interactions between water and the substrate. With regards to the water-surface terms (e.g., interactions between ice and surrounding liquid) in homogeneous ice nucleation, we agree with the reviewer that these terms could also complicate its ice nucleation process, and our findings of entropic stabilization could be applied to the homogeneous nucleation too. In fact, for homogeneous ice nucleation, the configurational entropy due to the stacking disorder of cubic and hexagonal ice has been reported in *Nature* 551, 218–222 (2017) and this configurational entropy term was shown to further stabilize the transition state of the homogeneous ice nucleation. Specifically, this previous study shows that the ice nucleus with

a mixture of cubic and hexagonal ice molecules lead to more configurations (higher entropy) than that of pure hexagonal ice, and thus facilitates the homogenous ice nucleation. In terms of energy, the interaction energies between cubic ice and surrounding water are shown to be comparable with those between hexagonal ice and surrounding water in the homogenous ice nucleation.

In the revised manuscript, we have included the discussion. Again, we would like to thank the reviewer for raising this insightful comment.

“...Similar mechanisms occur in homogeneous ice nucleation⁴⁸, where the critical ice nucleus is stabilized by the increase in entropy from the stacking-disordered cubic and hexagonal ice. However, unlike homogeneous ice nucleation, the classical HIN pathway, which forms mainly hexagonal ice, is also accessible with comparable flux due to the favourable potential energy for the critical nucleus (TS I).”

4. It would be wonderful if the authors could directly upload somewhere the input 230,240,250 K trajectories used in this work as it would help with reproducibility as well as help others applying different methods to the problem.

Reply: We would like to thank the reviewer for this good idea. We have followed the reviewer’s suggestion by uploading the initial MD trajectories in the Open Science Framework (DOI [10.17605/OSF.IO/YEURC](https://doi.org/10.17605/OSF.IO/YEURC)). We have included the link the in revised manuscript.

“The MD trajectories for $T=230$ K, 240 K, and 250 K used to build the MSMs have been deposited Open Science Framework (DOI [10.17605/OSF.IO/YEURC](https://doi.org/10.17605/OSF.IO/YEURC)).”

REVIEWERS' COMMENTS

Reviewer #1 (Remarks to the Author):

The authors have responded carefully and convincingly to my major concerns.

The major problem related to the 'apparent' stabilization of a large ice cluster was mainly a technical issue that the authors have clarified. The possible artifact due to truncation of the wall-liquid interactions has been ruled out.

My second main concern, the comparison of heterogeneous nucleation with homogeneous nucleation has been added and shows that heterogeneous nucleation remains a far more important pathway than homogeneous nucleation even at temperatures corresponding to 'critical homogeneous nucleation' (side note: with current uncertainties on nucleation rates, the statement that homogeneous nucleation has a rate of $10^{5.9299 \pm 0.6538}$ per s m³ seems somewhat too constraint. I find 10^6 per s m³ more than sufficiently accurate and likely also more consistent with current uncertainties on this problem).

I am not altogether persuaded by the mechanistic explanation for the two different kinetic path ways and the entropic effect, but I think this is a very interesting debate and the results are there for authors to consider different points of view.

In summary, the authors have responded satisfactorily to my main concerns. The paper is very interesting and carried out with great expertise. It provides a valuable insight on possible mechanisms for heterogeneous ice nucleation and a very interesting methodology for the study of activated events. I am now happy to recommend it for publication.

Dr. Luis G. MacDowell

Reviewer #3 (Remarks to the Author):

I am happy with this revised manuscript which has numerous extra analyses compared to the last round. I recommend publication as is.

Response to reviewer #1's comments:

The authors have responded carefully and convincingly to my major concerns.

The major problem related to the 'apparent' stabilization of a large ice cluster was mainly a technical issue that the authors have clarified. The possible artifact due to truncation of the wall-liquid interactions has been ruled out.

My second main concern, the comparison of heterogeneous nucleation with homogeneous nucleation has been added and shows that heterogeneous nucleation remains a far more important pathway than homogeneous nucleation even at temperatures corresponding to 'critical homogeneous nucleation' (side note: with current uncertainties on nucleation rates, the statement that homogeneous nucleation has a rate of $10^{(5.9299 \pm 0.6538)}$ per $s\ m^3$ seems somewhat too constraint. I find 10^6 per $s\ m^3$ more than sufficiently accurate and likely also more consistent with current uncertainties on this problem).

I am not altogether persuaded by the mechanistic explanation for the two different kinetic pathways and the entropic effect, but I think this is a very interesting debate and the results are there for authors to consider different points of view.

In summary, the authors have responded satisfactorily to my main concerns. The paper is very interesting and carried out with great expertise. It provides a valuable insight on possible mechanisms for heterogeneous ice nucleation and a very interesting methodology for the study of activated events. I am now happy to recommend it for publication.

Reply: We appreciate the reviewer's recognition of our work. We agree with the reviewer's suggestion to loosen the constraint on the rate of homogeneous ice nucleation.

To include the above point, we have updated the following sentence in the maintext of the revised manuscript:

"...which is orders of magnitude faster than the rate for homogeneous ice nucleation ($J_{hom} \approx 10^{6 \pm 1} s^{-1} m^{-3}$)⁵⁶."

Response to reviewer #3's comments:

I am happy with this revised manuscript which has numerous extra analyses compared to the last round. I recommend publication as is.

Reply: We thank the reviewer again for his/her recognition of our work.